# Mechanochemical Treatment in High-Shear Thermokinetic Mixer as an Alternative for Tire Recycling

**DOI:** 10.3390/polym14204419

**Published:** 2022-10-19

**Authors:** Otávio Bianchi, Patrícia Bereta Pereira, Carlos Arthur Ferreira

**Affiliations:** LAPOL/PPGE3M/Universidade Federal do Rio Grande do Sul, Av. Bento Gonçalves 9500, Porto Alegre 90010-150, Brazil

**Keywords:** ground tire rubber, mechanochemistry, recycling, thermokinetic mixer

## Abstract

This publication highlights the use of a high-speed thermokinetic mixer as an alternative to recycling ground tire rubber (GTR) using mechanochemical treatment. The GTR initially had a gelled fraction of 80% and presented a reduction of up to 50% of gel fraction in the most intensive condition (5145 rpm, n_2_). The processing condition at the lowest speed (2564 rpm, n_1_) resulted in greater selectivity in chain scission (*K*~1). However, in the most intense processing condition (10 min to n_2_), more significant degradation was observed via random scission, reduction in the glass transition temperature, T_g_ (11 °C), increase in the soluble polymeric fraction, and a more significant reduction in the density of bonds occurs. The artificial neural network could describe and correlate the thermal degradation profile with the processing conditions and the physicochemical characteristics of the GTR. The n_2_ velocity resulted in the formation of particles with a smoother and more continuous surface, which is related to the increase in the amount of soluble phase. The approach presented here represents an alternative to the mechanochemical treatment since it can reduce the crosslink density with selectivity and in short times (1–3 min).

## 1. Introduction

Car tires are the most relevant rubber items in terms of volume and importance, accounting for about 70% of the synthetic and natural rubber produced in the world. In general, they are composed of 40–50% elastomers such as raw rubber (NR, SBR, etc.), 20–30% fillers (carbon black, silica, etc.), 5–10% fibers and reinforcement (rayon, polyester, steel, etc.) and finally a tiny percentage of vulcanizing agents [1,2]. Unfortunately, the search for products with high performance, durability, and long life results in a polymer that is difficult to discard and complex to recycle [1].

At the end of the tire’s life cycle, its destination can be recovery, incineration, pyrolysis, or landfill. In incineration, waste tires are completely burned for energy generation in cement kilns and for producing steam, electricity, pulp, paper, lime, and steel [1,2]. In pyrolysis, the rubber component decomposes in the presence of heat and in the absence of oxygen, which prevents oxidation, and makes it possible to obtain pyrolysis oil, carbon black, and others [1,2]. Other approaches include the use of GTR in polymer blends. These mixtures are prepared with thermoplastics such as polyethylene (PE), polypropylene (PP), ethylene-vinyl acetate copolymer (EVA) [3], and polylactic acid (PLA) [4]. This approach allows for better-quality blends when devulcanization occurs [5]. In the devulcanization process, the focus is on selective scission, or in a way that allows the elastomer to be reprocessed. When scission is selective, the sulfide bonds undergo scission using specific conditions, which reduces the crosslinked fraction and often allows revulcanization [6]. Overall, these processes require rubber separation, and the wire and steel mesh are recovered as qualified iron scrap, and the nylon fabric is recovered and can be used as reinforcement in cardboard packaging [7].

Devulcanization or recovery methods have particularities, and their acceptability must be associated with the final destination of the treated product. Regarding devulcanization, the most efficient methods are the chemical method [8], the microwave-assisted method, the ultrasonic method, the biological method, the thermal method, and the mechanical method [7,8]. Mechanical processes are those in which the rubber is subjected to high shear stress forces [1,2,7,8]. Thermo-mechanical processes can be carried out continuously or in batches. In the continuous process, a twin-screw extruder is commonly used, in which the thread profile must be precisely adjusted to promote a good balance of bond breaks that allows the reuse of the elastomer. The batch process is usually performed in roller mixers or some other equipment specifically designed to generate the high levels of shear that benefit this process [1,2,6,8]. Furthermore, these methods can be aided by chemical agents to aid in processing conditions or chain breaking.

Ujianto and coworkers compared elastomer devulcanization using single- and twin-screw extruders. They reported that the twin-screw extruder operating at 300 °C and 100 rpm reduced the gel content by 50% for an elastomer with an initial gel content of 85%, while for the single-screw, the reduction was 40% [9]. Diaz et al. performed thermomechanical devulcanization of waste tires and compared two mixers, one with high shear and the other with pressure. In the study, the authors varied the temperature of the rubber and the mechanical energy consumed in operation, feeding the instrument with large pieces of rubber [10]. The machine reduced the particle size by rotating the rotor, thus increasing the content of soluble material in the rubber by 5% and decreasing its crosslink density by five times for the smallest particle size (100 μm) [10,11]. Studies show that using shear forces can reduce the particle size and the fraction of gelled material.

WO 2011/113148 deals with a method and equipment used to regenerate ground vulcanized rubber. The equipment consists of a thermokinetic mixer with a hermetic stationary chamber with a non-uniform internal surface, a rotor shaft that extends coaxially into the chamber, and a speed controller that varies up to at least about 2000 rpm. The method comprises pre-mixing the shredded rubber with oil at room temperature and then transferring the mixture to the thermokinetic mixer. The speed is increased until a devulcanization temperature is reached (300 to 330 °C for 0.25 to 3 s). After this condition, the speed is reduced until a new mixture temperature condition is reached (150 to 250 °C for 2 to 30 s). The best devulcanization condition leads to a 61% reduction in crosslink density. One of the advantages of this method is that there is no need to use auxiliary devulcanization chemicals, making this method ecologically correct or “green” [12].

Cavalieri and coworkers evaluated the effect of mechanochemical activation on the surface of solid-state devulcanized mid-tire rubber [13]. The authors show a reduction of 3% in the gel content and more than 50% in the crosslink density in 15 h of treatment in a ball mill. Interestingly, surface activation often allows the GTR to be reincorporated into fabrics with other elastomers and thermoplastics polymers more easily. Furthermore, during the GTR recovery process, shear forces increase friction between particles and exothermic reactions. Consequently, it can generate self-heating in the elastomer during its recovery [14].

Thermokinetic mixers can be used to generate heat and often melt the material. A crosslinked PU recovery approach was used by Gonela and coworkers, in which PU particles were depolymerized in a thermokinetic mixer. For the processing time of 90% to 30% reduction in gel content occurred [15]. Subsequently, the authors use PU after thermomechanical treatment to obtain blends with TPU and PA12. Other studies for the use of thermokinetic mixers deal with obtaining blends with crosslinked PE and EVA with high amounts (>60 wt.%) of crosslinked material [16,17]. Tire waste generally has a high gel content (>60%) and a more significant limitation for production if blended with thermoplastics or reused in industrial processes. In addition, high vulcanization limits adhesion between phases when used in mixtures and results in products with poor surface finish. Thus, thermokinetic mixers represent good opportunities to be used to treat crosslinked elastomers to improve and expand their use in the polymer and elastomers industry.

Thus, in this study, an alternative was proposed for the treatment of ground rubber from tires (GTR) using mechanochemical treatments using a high-speed mixer. Processing parameters that affect gel content, crosslink density, and particle morphology were investigated. In addition, the selectivity of chain scission was evaluated using the Horikx approach. Finally, an artificial neural network (ANN) was used to correlate the processing characteristics with the thermal degradation of the samples after mechanochemical treatment. For the system studied, it is noted that random splitting dominates the degradation process from processing times of 10 min at speeds of 5145 rpm (n_2_). Furthermore, the proposed treatment can be described through a percolation relationship that establishes that the chain’s scission results in the formation of the soluble fraction.

## 2. Materials and Methods

The ground tire rubber (GTR) with a particle size of 0.5–0.7 mm from the car, truck, tractors, and others, free of metallic parts, ground, and sieved, was supplied by a recycling industry located in the Rio Grande do Sul (Brazil). The composition of the GTR was evaluated by EDS and is shown in the Appendix A). Acetone P.A. (Neon, Suzano, Brazil), ethanol 92.8 INP (Neon, Suzano, Brazil), hexane P.A. (Neon, Brazil), and Toluene P.A. (Neon, Suzano, Brazil).

### 2.1. Mechanochemical Treatment

GRT samples containing 200 g were added to the mixing chamber of the DRAIS thermokinetic M.H. Equipment Ltd. (Guarulhos, Brazil) mixer model MH-50. This type of equipment heats the sample through the friction of the system and the material. In the mixer, it is possible to use two speeds in which its blades rotate at n_1_ = 2565 rpm and n_2_ = 5145 rpm. The mixing chamber was constantly cooled with circulating water at 25 °C. Different processing conditions were used at speeds 1 (n_1_) and 2 (n_2_), in which the time was varied (1–60 min). Immediately after processing, the temperature of the samples was read and cooled for further analysis.

### 2.2. Characterizations

The glass transition (T_g_) was determined using a TA Instruments DSC model Q 20 and some 6 mg of sample, and the temperature ranged from −80 to 250 °C, with a heating rate of 10 °C/min under an inert atmosphere (N_2_, 50 mL·min^−1^). Next, the thermal degradation profile and the filler content were determined with a TA Instruments TGA model Q 50. The analysis was performed in an inert atmosphere (nitrogen) and an oxidizing atmosphere after 600 °C for the determination of carbon black (synthetic air) (50 mL·min^−1^). The heating rate was 20 °C/min, from 30 °C to 900 °C, using ~10 mg.

The gel content was determined using about 7 g of samples using extraction (soxhlet) in acetone and hexane. Extraction with acetone was carried out for 16 h and in hexane for 24 h, then the samples were dried [18]. The gel content was considered the load fraction of the sample after the mechanochemical treatment. The density of the samples was determined according to ASTM D 792.

The crosslink density (Equation (1)) was determined according to the standard ASTM D6814 and using the Kraus correction (Equation (3)) [19]. The swelling was performed with 2–3 g) samples packed in paper bags. This determination was made on the samples after extraction in acetone and hexane. The equilibrium of swelling was reached after 20 days. Then, the excess solvent was removed from the sample’s surface, and the mass measurement was performed. The crosslink density was calculated according to:(1)ν=−[ln(1−Vr )+Vr+χ1Vr2 ]Vm1(Vr13−Vr )2,
where *ν* is the crosslink density per unit volume; *V_r_* is the polymer volume fraction in a swollen network in equilibrium with pure solvent; χ_1_ is the polymer–solvent interaction parameter (Flory–Huggins), and *V_m_*_1_ is the molar volume of the solvent.
(2)$$V_{r}=\frac{\left(\frac{m_{e}}{\rho_{b}}\right)}{\left(\frac{m_{e}}{\rho_{b}}\;\right)+\left(\frac{m_{s}}{\rho_{s}}\right)},$$
where *m_e_* is the mass of extracted rubber, *ρ_b_* is the density of the extracted rubber, *m_s_* is the mass of solvent adsorbed by the sample, and *ρ_s_* is the density of the solvent. The interaction parameter χ_1_ used in this work was considered 0.391 at 25 °C for Cis-polyisoprene-toluene [20].
(3)$$\frac{V_{r}}{V_{R}}=1-\frac{\Phi \left[3c\;\left(1-\sqrt[{3}]{V_{r}}\right)+V_{r}-1\right]}{1-\;\Phi },$$
where *V_r_* is calculated according to Equation (4), *V_R_* is the volume fraction of polymer in a swollen network in equilibrium with charge-corrected pure solvent. Φ is the volumetric fraction of the filler in the sample, determined by TGA, and calculated according to Equation (4), and *c* is a constant relative to the fillers present in the solvent-independent composition (for fillers such as carbon black, the value is 1.17) [19,21].
(4)$$\Phi =\frac{m_{\;}{}_{c}\times \rho_{g}\times m_{i\;}}{\rho_{c}\times m_{f}},$$
*m_c_* is the mass of fillers, *m_i_* is the mass of the polymer before swelling, *m_f_* is the mass of the polymer after swelling, *ρ_g_* is the density of the polymer, and *ρ_c_* is the density of the fillers. The efficiency of the mechanochemical treatment, *X* was made by comparing the crosslink densities before and after the treatment.
(5)$$X=\left[1-\left(\frac{\nu }{\nu_{c}}\right)\right]\times 100,$$
where *ν* is the crosslink density per unit volume and *ν_c_* is the crosslink density per unit volume of the sample without mechanochemical treatment.

The theoretical relationship between the soluble fraction generated after the degradation of a polymeric network and the relative decrease in crosslink density as a result of main-chain scission or crosslink scission [21,22] was based on Horikx’s approach, using the Flory–Huggins theory as a basis [23]. However, when the polymer is degraded through the statistically random distribution of chain scission events, Equation (6) is adopted. In this scenario, where the degradation consists only of the crosslinks’ selective scission, strictly corresponding to an ideal devulcanization process, the equation used is the 7 [22].
(6)$$1-\frac{v_{f}}{v_{i}}=1-\frac{{\left(1-\sqrt{s_{f}}\right)}^{2}}{{\left(1-\sqrt{s_{i}}\right)}^{2}},$$
(7)$$1-\frac{v_{f}}{v_{i}}=1-\frac{\gamma_{f}{\left(1-\sqrt{s_{f}}\right)}^{2}}{\gamma_{i}{\left(1-\sqrt{s_{i}}\right)}^{2}},$$
where *ν_i_* and *ν_f_* are the crosslink densities in mol·cm^−3^; $s_{i}$ and $s_{f}$ the sol fractions. $\gamma_{i}$ and $\gamma_{f}$ are the crosslinking indices, computed according to Equation (8). The subscripts “*i*” and “*f*” represent the samples before and after devulcanization, respectively [21,22].
(8)$$\gamma =v_{x}\frac{M_{n}}{\rho },$$
where *ρ* is the density in g·cm^−3^ of the polymer, $v_{x}$ is the crosslink densities in mol·cm^−3^, *M_n_* is the original polymer chains’ average molecular weight (g·mol^−1^) before the crosslink formation.

It was proposed to correlate these characteristics with the thermal degradation profile (TGA) using an artificial neural network (ANN) after the mechanochemical treatment from the assumption that there is a reduction in the crosslink density and gel content. The ANN was conventionally constructed with three layers: an input layer, an output layer, and a hidden layer. Each layer has a different number of neural elements. In the present study, information from TGA curves, speed of mechanochemical treatment, processing time, gel content, and bond density were input vectors. Thus, the generated network will modify the weight of the interconnections between neurons to reproduce the parameters provided to obtain the mass loss curves. Figure 1 shows a schematic of the network used.

In this procedure, the initial information to be considered includes the number of initial TGA curves and other characterizations of the GTR after the mechanochemical treatment. With network training, it is possible to predict new TGA curves between the lower and upper ranges and obtain information about new experimental conditions to avoid accumulating errors. The ANN fit was created with 4 layers, 20 hidden neurons in each layer, k-fold cross validation = 5, error summation squared error function, activation function = ReLU, learning rate = 0.001 and loss tolerance = 1 × 10^−6^. The algorithm used was resilient backpropagation with tracking.

Fourier transform infrared spectroscopy (FTIR) was used to evaluate the possible degradation of the GTR after mechanochemical processing. Measurements were made with the soluble fraction in Perkin Elmer equipment, model 1000 in KBr pellets (4000−400 cm^−1^, 32 scans, and 4 cm^−1^ of the resolution).

The morphology of the samples was investigated by scanning electron microscopy (FEG-SEM, Tescan MIRA3, Czech Republic) with a dispersive X-ray spectroscopy (EDS) detector (Oxford Instruments). The samples were fixed on carbon tapes, and a thin layer of gold was deposited by sputtering.

## 3. Results and Discussion

Table 1 shows the temperature, acetone, and hexane soluble fractions for GTR samples before and after mechanochemical treatment. As the processing time was increased, the temperature increased proportionally from 32 °C for 1 min in n_1_ to 68 °C in 60 min. This result is expected since the thermokinetic mixer generates heat by friction. However, as the speed has practically doubled, a much more significant increase in temperature is noted, reaching 185 °C in 10 min for n_2_. Although, the soluble fractions presented constant values in n_1_ and 1 min processing in n_2_, an increase in these fractions was later noticed. This increase in the soluble fraction may be related to chain scission that produces smaller fragments that can be extracted.

The gel fraction (*ξ*) and crosslink density (*ν*) are important parameters to characterize the efficiency of the mechanochemical process of GTR recovery. However, the gel fraction is less sensitive to structural changes. Therefore, it is often possible to activate the surface of the GTR particle with faster treatments and reincorporate it into a production process. Although some approaches, such as using ball mills, produce exciting results [13], the thermokinetic mixer is capable of generating high shear rates [24] (>10^4^ s^−1^), and it plays an essential role in reducing the crosslink density. On the other hand, at higher shear rates, the oxidation of the elastomeric compound may increase [14]. Thus, depending on which application the recovered elastomer will be used, the best processing conditions for recovery will be defined.

Table 2 shows the results of the gel fraction, crosslink density, and mechanochemical treatment efficiency for samples treated under different conditions in a thermokinetic mixer. Here, the gel fraction was estimated after extractions in acetone and hexene. The gel content is less sensitive to the mechanochemical treatment since some crosslinks on the surface are initially broken, and only when the medium is more aggressive is there a chain thermomechanical degradation by random scission, which results in changes in the gel content. In the crosslink density, it can be verified that they already produce an efficiency of approximately 50% in milder processing conditions. These results show that depending on where the elastomer will be used, as in mixtures with other elastomers or thermoplastics, there may not be a need for further degradation of the polymer chain, as reported by Cavalieri and coworkers [8].

Ujianto et al. [9], using twin-screw extruders, reached gel values of approximately 50%, from samples with 85% gel content at 300 °C and 100 rpm. Considering that the speed used in extrusion is 100 rpm and that in the thermokinetic mixer, the degree of devulcanization is similar. The shear rate is believed to be essential in reducing the gel content, as the rotor speeds are about 50 times higher in the thermokinetic mixer.

On the other hand, when the temperature increases due to friction, it can result in greater chain scissions that will reflect in the reduction in gel content and increase in density. For example, Simon, and coworkers [22,23] compared the devulcanization of the GTR in a laboratory microwave oven and an internal mixer. For a sample treated at 120 rpm in the mixer, which reached a temperature of 200 °C, the sol content was 30.2% by weight, and the crosslink density was 3.7 × 10^−4^ mol·cm^−3^, while a sample treated at 40 rpm, which reached a temperature of 160 °C, the sol content was 14.4% and the crosslink density was 7.3 × 10^−4^ mol·cm^−3^. Therefore, a more significant reduction in the gelled fraction and crosslink density is noticed when the system reaches higher temperatures. However, thermomechanical degradation favors the reduction in the polymer fraction as light fractions of degradation products are evaporated.

The DSC results not shown here show only a glass transition (T_g_) for the GTR at −58.9 °C, which agrees with the literature [25,26]. Note that there is no variation in the processing times of 1 and 3 min at speed n_1_. On the other hand, when GTR was processed for 10 min in n_2_ there was a reduction to −69.8 °C. Thus, when there is a reduction in T_g_, there is an increase in the segmental mobility of the chains. This may be related to the possible scission of chains under more severe shear conditions for a longer time.

Figure 2a illustrates the TGA curves for the GTR sample after processing in a thermokinetic mixer. Here, it was chosen to present the conditions in 10 min in n_1_, 60 min in n_1_, and 10 min in n_2_. Along with the experimental data, the neural network’s prediction for the behavior of the thermal degradation profile of the samples and an experimental prediction is being shown. All samples presented the temperature of the beginning of thermal degradation ~250 °C, with maximum rates around 500 °C. Due to the GTR being formed by different elastomers, the thermal loss mass profile occurs in multiple steps, as can be seen for a maximum around 350 °C which can be related to natural rubber, and at 470 °C it is related to styrene–butadiene rubber, which is the main elastomers used in the manufacture of automobile tires [1,7]. In addition, chain scission forms molecules of a lower molecular mass composed of hydrocarbon species (-CH_2_-) and CO_2_ [1]. For the samples processed in n_1_, a carbon black content of around 31 wt.% was found. However, when they were processed in n_2_ a change was noticed, and this amount increased to 36 wt.% as well as the other relative amounts of fillers (silica and others). This change in the mass loss profile is associated with the fact that during the mechanochemical process, chain scission and volatilization of low molecular mass compounds occur. As a result, there is a change in the relative fraction of carbon black and inorganic compounds.

The ANN was used to predict the observed behavior of the complex mass loss profile. First, experimental data were compared with predicted theoretical data, as shown in Figure 2b, and a good agreement R^2^ > 0.999 was noted. Subsequently, it was possible to predict the experimental data for other conditions (blue curve, Figure 2a). It is also noted that the cross-validation (Figure 2c) results in low dispersion.

The selectivity of the scission of chains generated by the mechanochemical process in a thermokinetic mixer, the Horikx method, was used, which allows distinguishing whether the scission of chains tends to be more selective or random. For this analysis, it is necessary to carry out a mass balance in the material since the filler and polymer fractions are altered during the treatment.

The theoretical lines in Figure 3 were constructed using Equations (6) and (7), respectively, varying the fraction of solubles from 0.10 to 0.99 concerning the decrease in crosslink density. The crosslinking index was obtained through Equation (8), considering the numerical average molecular weight of the NR polymer of 200,000 g·mol^−1^, according to the literature [21,22]. Based on the proximity of the curve (random or selective), it is possible to infer whether the dominant phenomenon during the mechanochemical process is more selective or random [21,27]. Selectivity can be achieved using a coefficient (*K*). Two possible approaches are to set the selectivity parameter towards the soluble fraction (vertical) or the crosslink density (horizontal). The selectivity parameter of the soluble fraction (*K_s_*) can be calculated by Equation (9) [21].
(9)$$K_{s}=\frac{S_{c}-S}{S_{c}-S_{x}},$$
where *S_c_* is the theoretical sol fraction for random scission, *S_x_* is the theoretical sol fraction for selective crosslinking scission, and *S* is the measured soluble fraction. The selectivity parameter in the crosslink density direction (*K_x_*) can be calculated by Equation (10) [18].
(10)$$K_{x}=\frac{X-X_{c}}{X_{x}-X_{c}},$$
where *X_c_* is the theoretical values related to the relative decrease in crosslink density for random scission, and *X_x_* is the theoretical values of the relative decrease in crosslink density for selective scission. The general selectivity parameter (*K*) is then defined according to Equation (11) [21].
(11)$$K=\frac{K_{s}+K_{x}}{2},$$

Thus, when random chain scission is dominant, *K* = 0, and when selective *K* = 1.

The soluble fraction showed a practically linear variation for the experiments at n_1_ ($S=0.20+1.46\times 10^{-4}*t,\;R>0.98)$ and n_2_ (S=0.19+0.02∗t, R>0.98) in relation to the processing time. Thus, the gel fraction changes much more intensely when shear is increased. Regarding the decrease in crosslinks ($1-\frac{\nu_{f}}{\nu_{i}}$) can be fitted by logistic equations in relation to the processing time. For n_1_, $1-\frac{\nu_{f}}{\nu_{i}}=0.49+\frac{6.69\times 10^{-5}-0.49}{1+{\left(\frac{t}{t_{0}}\right)}^{1.53}}$, R > 0.98, and for n_2_, $1-\frac{\nu_{f}}{\nu_{i}}=3.30+\frac{1.50\times 10^{-5}-3.30}{1+{\left(\frac{t}{t_{0}}\right)}^{0.08}}$, R > 0.99. This observation may be related to the reduction in crosslinks following a sigmoidal process [28]. That is, there is an initial period plus reading, a period of acceleration, and a limit is reached for the processing technique employed in the mechanochemical treatment.

Figure 3 shows the results of the Horikx analysis for the two speeds (n_1_ and n_2_) and times of mechanochemical processing. The blue line illustrates the region in which the process is selective, while the red line illustrates the random scission. A comparison was also made with the data generated by ANN. All samples processed at n_1_ show selective chain scission with *K* = 0.98–1) and at 1 min in s_2_. Subsequently, there is an increase in thermomechanical degradation by the random scission in the mechanochemical treatment with K, going from 0.80 to 0.39 in the 10 min in n_2_ condition, which indicates that the high shear induces the scission of the C–C bonds of the backbone chain.

Edwards and coworkers [21] used Horikx analysis to indicate the selectivity of crosslink scission during the devulcanization of GTR by mechanical shear at elevated temperatures in a twin-screw extruder. The extruder used operated at 30–80 rpm and 175–275 °C. The best selectivity condition found by the authors was obtained in the sample submitted to 275 °C and 55 rpm. This best shear condition is about forty-six times lower than the best condition in this study. Furthermore, the working temperature was five times higher than the final temperature of the best condition evaluated by Horikx in the study in the thermokinetic mixer. Simon et al. [27,29], compared the thermomechanical treatment in a mixing chamber (Brabender Plasti-Corder) with microwave processes. They noticed that chain scissions are more selective in the internal mixer. However, when there is an increase in temperature, there is a tendency for the polymer chain to degrade, as reported by the same authors. The literature [11] reports that the devulcanization process will occur extremely slowly if the operating temperature is below 50 °C. This observation is in line with the data presented in this study, in which similar conditions and greater selectivity are achieved in n_1_.

Figure 4 shows that the normalized gel fraction (*ξ*/*ξ*_0_) as a function of the normalized crosslink density (*ν*/*ν*_0_). In this Figure the experimental data was fitted with the empirical model proposed by Isayev, according to Equation (12) [30]:(12)$$\frac{\xi }{\xi_{0}}=\left(1+\psi ln\frac{\nu }{\nu_{0}}\right)H\left[\frac{\nu }{\nu_{0}}-exp\left(-\frac{1}{\psi }\right)\right],$$
where *H* is the Heavi-side step function because the gel fraction is always positive and negative values and has no physical meaning, such that $1+\psi ln\frac{\nu }{\nu_{0}}>0\;\mathrm{if}\;\frac{\nu }{\nu_{0}}-exp\left(-\frac{1}{\psi }\right)>0$. The parameter *ψ* represents the scales of the relative change in gel fraction concerning the change in the initial crosslink density, *ν*_0_ should be controlled by the relative to chain scission probability.

The empirical model (Equation (12)) was able to describe all data in a universal curve. Thus, using the same assumptions as Isayev [30], it is possible to infer that the mechanochemical treatment can be described by a network percolation model, in which the ratio between the amounts of intermolecular bond breakage and main chain bond breakage resulting in the formation of the fraction soluble [30,31]. This model implies that devulcanization and degradation coincide at random throughout the rubber network, a condition believed to be sufficiently satisfied during processing in a thermokinetic mixer. The value of the parameter ψ found was 0.13 ± 0.05, which is in agreement with the values found by Isayev [30] for ultrasonic devulcanization of GTR (ψ=0.17).

Figure 5 shows the FTIR spectra for the hexane-extracted GTR samples. In the 2950 and 2853 cm^−1^, the asymmetric and symmetrical stretching bands of the C-H bonds present in NR methyl groups are observed. At 1450 and 1375 cm^−1^ it is possible to observe the symmetrical deformation bands in the CH_2_ and CH_3_ planes and the cis-isoprene bands, around 885 cm^−1^, 1370 cm^−1^, and 1630 cm^−1^. At 700 cm^−1^ and 775 cm^−1^ characteristic bands of SBR were found, and at 965 cm^−1^ to butadiene [32,33]. The band that characterizes the vibration absorption related to the S-S bond in the GTR is found close to 462 cm^−1^ and can reach 495 cm^−1^ [33]. Regarding the effect of processing time and speed, it is noted that the region between 2950 and 2853 cm^−1^ increases proportionally with the intensity of the mechanochemical treatment.

The intensive mechanochemical treatment results in degradation and almost always oxidation. The oxidative process can result in the formation of new crosslinks, but it is counterbalanced by the high shear that results in chain scission, as noted by the samples processed in n_2_. The relative increase in bands at 1800 to 1500 cm^−1^ can relate to this effect (C=C and C=O), as well as the appearance of bands at 1500 to 750 cm^−1^ (C-O, S-O-C, C-O-C, S=O, C-C) [32,33,34].

Figure 6a–f illustrates SEM micrographs of the GTR samples before (Figure 6a,b) and after treatment in the thermokinetic mixer. The particles have elongated sizes, typical of a non-cryogenic milling process. Sizes range from 111 μm to 1345 μm, with a higher incidence of particles larger than 400 μm. The EDX analyses in the Appendix A show the presence of carbon, sulfur, silicon, aluminum, calcium, oxygen, zinc, and sodium. These elements are part of the compounds commonly used to manufacture tires [7]. After 3 min and 60 min of processing at n_1_, it is noted that the beginning of the fragmentation of the sample occurs (Figure 6c,d). The increase in the processing speed (n_2_) resulted in the formation of a sample with a smoother and more continuous surface. This characteristic is related to the increase in the amount of soluble phase. These results are similar to those found in the literature [33] using mechanochemical treatment in extrusion. As the conditions are intensified, the rubber particles present smooth contours.

## 4. Conclusions

GTR was submitted to mechanochemical treatment in a thermokinetic mixer under different processing conditions and experimentally characterized by physical, chemical, thermal, and morphological methods. Experimental techniques allowed for establishing relationships with processing conditions. Models based on an artificial neural network (ANN), Horikx, and Isayev analysis were used to describe how the different processing conditions affect the mechanochemical process. The most aggressive processing condition resulted in a reduction of 11 °C in the glass transition temperature. The crosslink density, as well as the gel content, respond directly to the effect in the processing conditions. That is, the more intense, the smaller the number of crosslinks and the greater the soluble fraction. Horikx analysis indicates that samples processed in s_1_ showed the highest concentration of selective chain scissions. However, when the thermokinetic mixer speed is increased, there is a greater tendency for random scission. The highest efficiency in the mechanochemical treatment was 53%, with one minute of processing in n_2_. The artificial neural network (ANN) was able to predict thermal degradation behavior and correlate with the mechanochemical treatment in a thermokinetic mixer. Both chain scission and devulcanization are processes that coincide, when chain scission increases, materials with smoother and more continuous surface textures are formed.

We conclude that the GTR produced by the thermokinetic process in high-speed mixtures has excellent potential for use due to its short processing time and efficiency in terms of selectivity. This mechanochemical treatment technique can increase the use of GTR in mixtures with elastomers and thermoplastics. Combining shear and specific conditions allows for a “green” and sustainable mechanochemical approach to the recovery of waste tires. The technique discussed here represents an exciting alternative to contribute to the circularity and reuse of these materials.

## Figures and Tables

**Figure 1 polymers-14-04419-f001:**
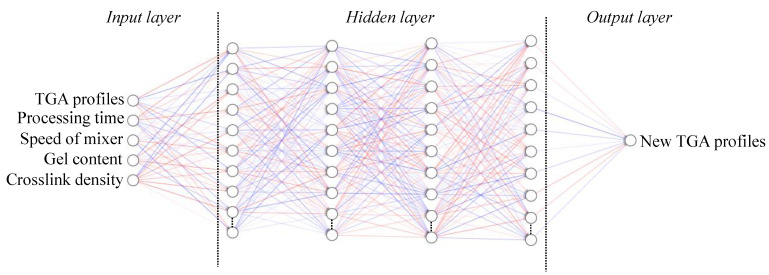
Scheme of the ANN used.

**Figure 2 polymers-14-04419-f002:**
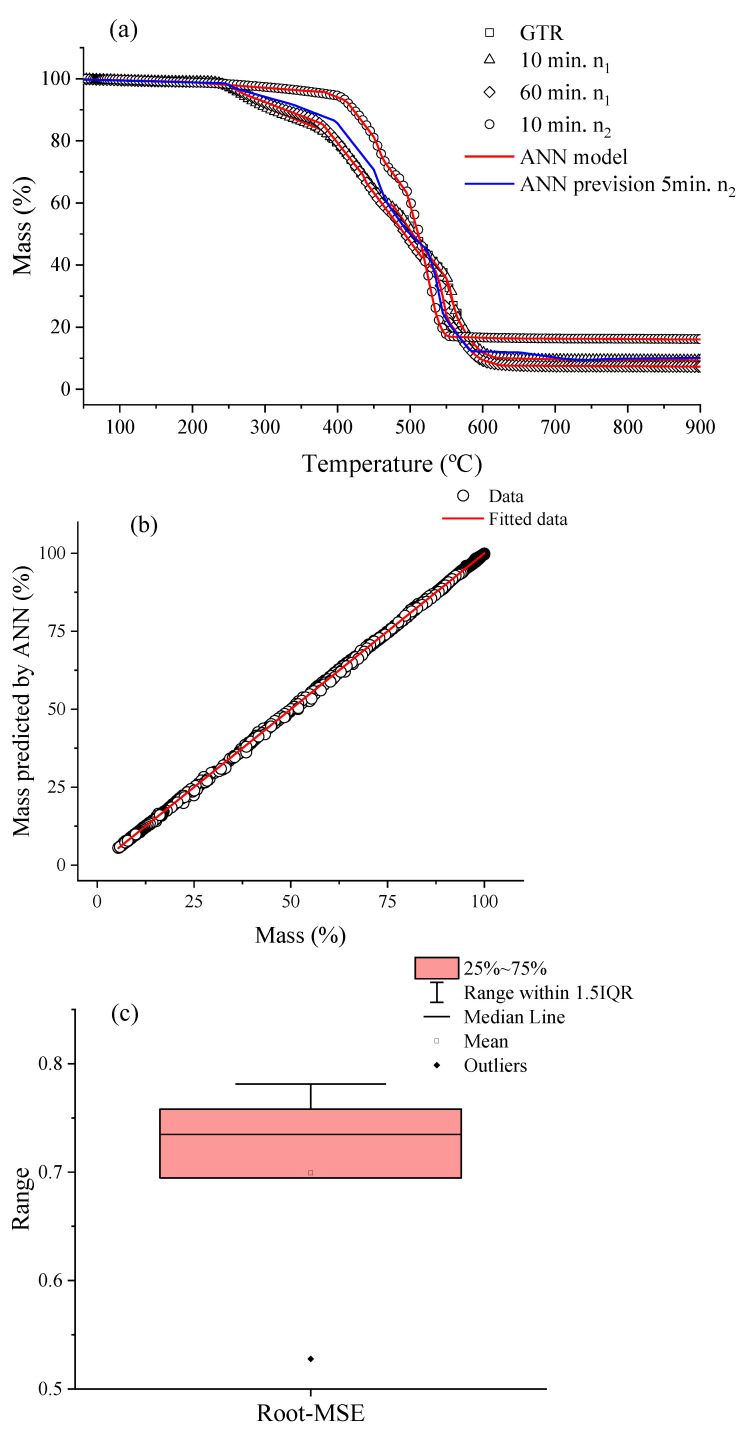
(**a**) TGA curves (20 °C·min^−1^) for GTR samples in different treatment times and speeds. The open dots symbols refer to experimental data, the solid red line to the ANN predicted data, and the solid blue line to a 5 min predicted condition in n_2_. (**b**) Comparison of the data fitted by ANN and experimental data and (**c**) Cross-validation box plot, R^2^ > 0.9999.

**Figure 3 polymers-14-04419-f003:**
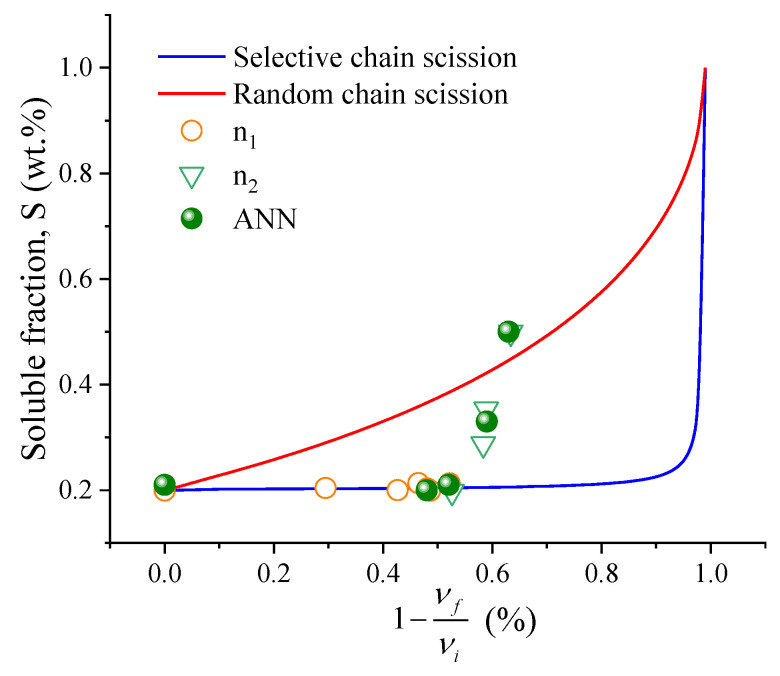
Soluble fraction as a function of the decrease in crosslinks ($1-\frac{\nu_{f}}{\nu_{i}}$) (Horikx analysis). The red line represents the theoretical random scission curve, the blue line selective scission, and ANN-generated green dots.

**Figure 4 polymers-14-04419-f004:**
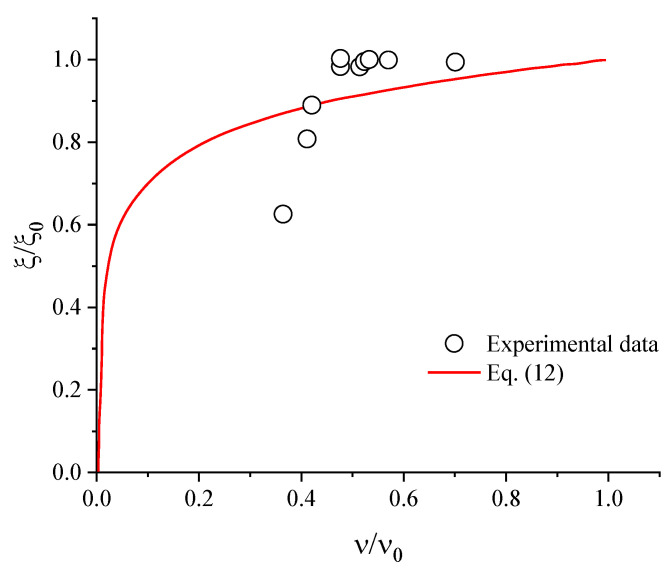
Normalized gel fraction versus normalized crosslink density for GTR processed in a thermokinetic mixer. Symbols are experimental data, and the solid line in red is the fit of Equation (12).

**Figure 5 polymers-14-04419-f005:**
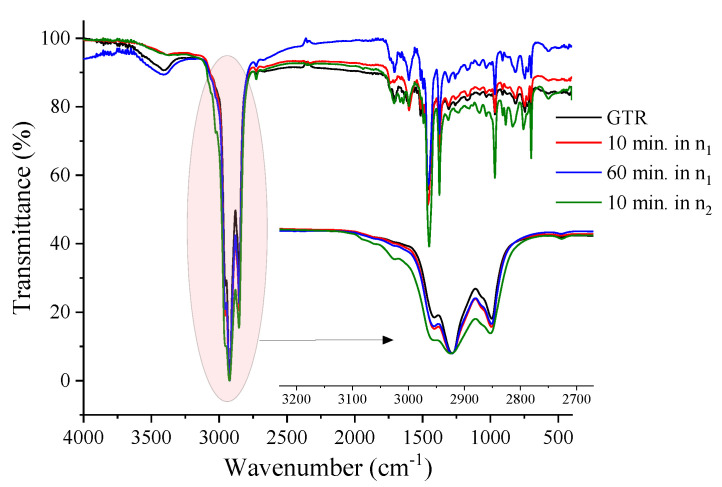
FTIR spectra for the soluble fraction of GTR after mechanochemical treatment.

**Figure 6 polymers-14-04419-f006:**
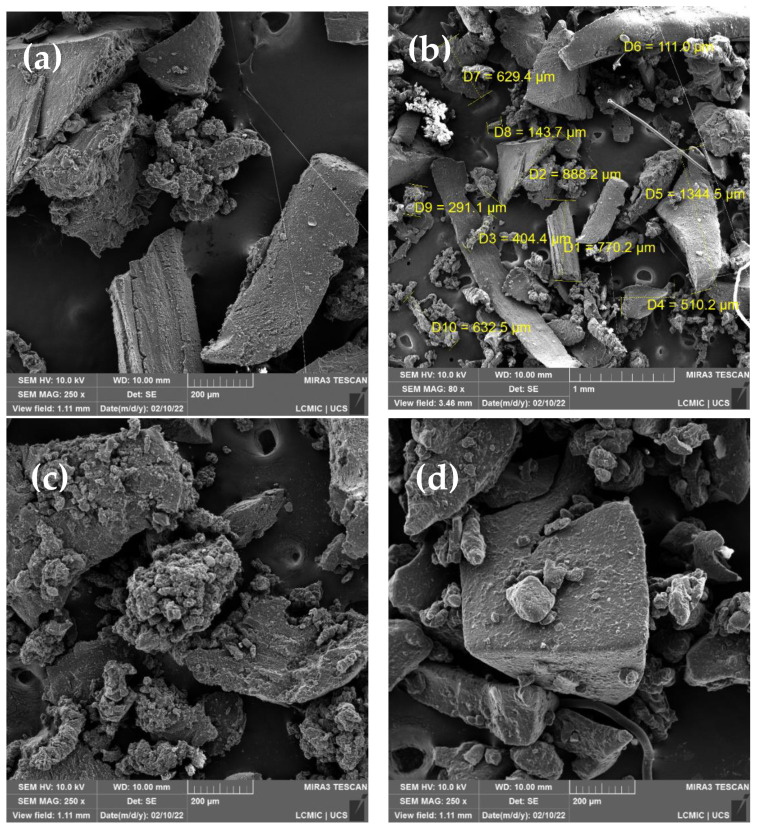
SEM micrographs for GTR: before mechanical treatment (**a**,**b**); after treatment at 3 min. (**c**) and 60 min. (**d**) in n_1_; 6 min. (**e**) and 10 min. (**f**) in n_2_.

**Table 1 polymers-14-04419-t001:** Processing conditions, the temperature at final processing, and solubles fractions for GTR after and before mechanochemical treatment.

Processing Speed	Time (min)	Temperature (°C)	Solubles Fraction in Acetone	Solubles Fraction in Hexane
-	-	-	0.11	0.12
n_1_	1	32	0.11	0.12
n_1_	3	43	0.11	0.12
n_1_	6	59	0.11	0.12
n_1_	10	54	0.11	0.12
n_1_	30	38	0.10	0.13
n_1_	60	68	0.10	0.13
n_2_	1	51	0.11	0.12
n_2_	3	169	0.13	0.16
n_2_	6	179	0.14	0.19
n_2_	10	185	0.18	0.27

n_1_ = 2565 rpm and n_2_ = 5145 rpm.

**Table 2 polymers-14-04419-t002:** Gel fraction (*ξ*), crosslink density (*ν*) and mechanochemical efficiency, *X*.

Processing Speed	Time (min)	Gel Fraction (wt.%)	Crosslink Density (mol·cm^−3^)	*X* (%)
-	-	77.3	0.00107	-
n_1_	1	77.4	0.00075	29
n_1_	3	77.1	0.00061	43
n_1_	6	77.1	0.00055	49
n_1_	10	77.2	0.00056	48
n_1_	30	77.1	0.00057	46
n_1_	60	77.1	0.00051	52
n_2_	1	77.3	0.00051	53
n_2_	3	70.5	0.00045	58
n_2_	6	66.5	0.00044	59
n_2_	10	55.7	0.00039	63

## Data Availability

The raw data needed to reproduce these findings can be shared by the authors upon reasonable request.

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
