# Peer review of "Mechanochemical Treatment in High-Shear Thermokinetic Mixer as an Alternative for Tire Recycling"

_polymers, 2022, doi:10.3390/polym14204419_

Round 1
Reviewer 1 Report
1. I have just a minor comment that is; Add one more keyword.
Author Response
Dear reviewer, we appreciate your comments. His suggestion has been included.
Reviewer 2 Report
This present work highlights the effect of mechanochemical treatment in a high-speed thermokinetic mixer as an alternative for recycling ground tire rubber (GTR). The artificial neural network could describe and correlate the thermal degradation profile and processing conditions with the physicochemical characteristics of the GTR. The approach presented here represents an alternative for mechanochemical treatment since it can reduce the crosslink density with selectivity and in short times.
The topic is interesting as it deals with a wide area of rubber applications. It falls within the esteemed journal of Polymers. The logic of the article is well organized, and the language is good. There are many experiments and equations in the work, and the result is incredible. It can be considered for publication after some modifications.
Minor points:
1. There should be a blank between the number and its following unit, such as 0.7 mm.
2. In Section 2.1, the parameters s1 and s2 are the angular velocities, and normally they are expressed as n1 and n2. At least they should be defined at first.
3. In Eq. (1), the biggest bracket is not necessary. Similar problem exists in Eq. (2).
4. Please check the sentence in L135.
5. After each equation, there should be “,” or “.”.
6. How about the shearing process? There is no mechanics analysis, and even there no related equations in the present work.
7. Please check the sentence in L246.
8. In Eq. (10), “X” is defined repeatedly.
9. In L321, the authors should notice that the units in the fitted formulas are not consistent on both sides.
Author Response
This present work highlights the effect of mechanochemical treatment in a high-speed thermokinetic mixer as an alternative for recycling ground tire rubber (GTR). The artificial neural network could describe and correlate the thermal degradation profile and processing conditions with the physicochemical characteristics of the GTR. The approach presented here represents an alternative for mechanochemical treatment since it can reduce the crosslink density with selectivity and in short times.
The topic is interesting as it deals with a wide area of rubber applications. It falls within the esteemed journal of Polymers. The logic of the article is well organized, and the language is good. There are many experiments and equations in the work, and the result is incredible. It can be considered for publication after some modifications.
Answer to reviewers: We appreciate the comments made by the reviewer. Toda as considerações serão revistas. All considerations will be reviewed.
Minor points:
- There should be a blank between the number and its following unit, such as 0.7 mm.
Answer to reviewers: Thanks for the comment. We have reviewed all spaces in the units.
- In Section 2.1, the parameters s1 and s2 are the angular velocities, and normally they are expressed as n1 and n2. At least they should be defined at first.
Answer to reviewers: Thanks for the comment. We adopted the nomenclature suggested by the reviewer and made adjustments to the text.
- In Eq. (1), the biggest bracket is not necessary. Similar problem exists in Eq. (2).
Answer to reviewers: Thanks for the comment. The equations have been corrected according to the suggestions.
- Please check the sentence in L135.
Answer to reviewers: The sentence has been corrected.
- After each equation, there should be “,” or “.”.
Answer to reviewers: The text has been corrected.
- How about the shearing process? There is no mechanics analysis, and even there no related equations in the present work.
Answer to reviewers: Thanks for the comment. In the work, we seek to evaluate processing conditions' effect on physical-chemical properties. The percolation model was able to describe the results well. However, to accurately estimate the shear rates in the shear chamber, it is necessary to use computional fluid dynamics (CFD) since there are regions where the stresses differ. The publication, DOI: 10.1002/app.21597, deals with a procedure for estimating shear rates in similar mixers. Nevertheless, indentation effects or agitator profiles must be considered.
To obtain a critical shear rate correlation for a process like this, we must vary the speeds, which the mixer we used does not allow. This may be an aspect of a future publication.
- Please check the sentence in L246.
Answer to reviewers: The text has been corrected.
- In Eq. (10), “X” is defined repeatedly.
Answer to reviewers: The text has been corrected.
- In L321, the authors should notice that the units in the fitted formulas are not consistent on both sides.
Answer to reviewers: The text has been corrected.
Reviewer 3 Report
Review on “Mechanochemical treatment in high-shear thermokinetic mixer an alternative for tire recycling”
by Bianchi et al.
Manuscript ID polymers-1981177
A- General Comments
The paper in hand concerns highlights on the effect of mechanochemical treatment in a high-speed thermokinetic mixer as an alternative for recycling ground tire rubber (GTR). The GTR initially has an 80% gelled fraction and presented up to 50% gel fraction in the most intensive condition (5145 rpm, 9s2). Particularly, it was shown by the authors that processing at the lowest speed (2564 rpm, s1) resulted in greater selectivity concerning the mechanochemical treatment (K=1). In the most intense processing condition (10 min. at s2), more significant degradation was observed via random scission with a reduction of the glass transition temperature, Tg (11ºC).
The topic of the paper is interesting, within the scope of the journal, and worthy of investigation. The originality of the work is acceptable and the study performed is adequate. However, the manuscript deserves a major revision. I suggest that authors take into account the comments and questions below before it can be accepted for publication in Polymers.
B- Detailed Comments and questions
Title
I guess a “as” should be added before “an alternative”.
Abstract
1- Results should be consistently presented in the abstract: the most important outputs only.
2- How the study was performed and results obtained should be explicated.
3- It is not clear whether the originality resides in the concept of mechanochemical treatment as an alternative or the experimental setup and study performed to test the effect of mechanochemical treatment. Please clarify
Keywords
Keywords are ok. GTR should be written in complete words and not abbreviations.
1- Introduction
1- References relevant to Polymers should be added, if possible.
2- There is a very good literature review presented. However, the originality of the work should be highlighted further at the end of the introduction.
2- Materials and Methods
1- This section is very well presented and complete. However, quality of figure 1 should be enhanced and references to most of the equations presented should be provided.
3- Results and discussion
1- There are a lot of interesting observations without deep analysis. More physical analysis are to be added to this section;
2- Quality of Figure 1 should be enhanced.
3- Table 1 at the end of page 6 should be Table 2.
4. Figure 2 is missing. Please do the necessary updates.
4- Conclusions
The main outputs of the manuscript in terms of applications should be highlighted.
5- References
References relevant to Polymers should be added, if possible.
Author Response
A- General Comments
The paper in hand concerns highlights on the effect of mechanochemical treatment in a high-speed thermokinetic mixer as an alternative for recycling ground tire rubber (GTR). The GTR initially has an 80% gelled fraction and presented up to 50% gel fraction in the most intensive condition (5145 rpm, 9s2). Particularly, it was shown by the authors that processing at the lowest speed (2564 rpm, s1) resulted in greater selectivity concerning the mechanochemical treatment (K=1). In the most intense processing condition (10 min. at s2), more significant degradation was observed via random scission with a reduction of the glass transition temperature, Tg (11ºC).
The topic of the paper is interesting, within the scope of the journal, and worthy of investigation. The originality of the work is acceptable and the study performed is adequate. However, the manuscript deserves a major revision. I suggest that authors take into account the comments and questions below before it can be accepted for publication in Polymers.
B- Detailed Comments and questions
Title
I guess a “as” should be added before “an alternative”.
Answer to reviewers: Thanks for the comment. the title has been changed
Abstract
1- Results should be consistently presented in the abstract: the most important outputs only.
2- How the study was performed and results obtained should be explicated.
3- It is not clear whether the originality resides in the concept of mechanochemical treatment as an alternative or the experimental setup and study performed to test the effect of mechanochemical treatment. Please clarify
Answer to reviewers: Thanks for the comment. The abstract has been rewritten
Keywords
Keywords are ok. GTR should be written in complete words and not abbreviations.
Answer to reviewers: Thanks for the comment. The term has been corrected.
1- Introduction
1- References relevant to Polymers should be added, if possible.
2- There is a very good literature review presented. However, the originality of the work should be highlighted further at the end of the introduction.
Answer to reviewers: The introduction has been revised and new references added.
- Colom, X.; Cañavate, J.; Carrillo-Navarrete, F. Towards Circular Economy by the Valorization of Different Waste Subproducts through Their Incorporation in Composite Materials: Ground Tire Rubber and Chicken Feathers. Polymers 2022, 14, 1090.
- Candau, N.; Oguz, O.; León Albiter, N.; Förster, G.; Maspoch, M.L. Poly (Lactic Acid)/Ground Tire Rubber Blends Using Peroxide Vulcanization. Polymers 2021, 13, 1496.
- Zedler, Ł.; Klein, M.; Saeb, M.R.; Colom, X.; Cañavate, J.; Formela, K. Synergistic Effects of Bitumen Plasticization and Microwave Treatment on Short-Term Devulcanization of Ground Tire Rubber. Polymers 2018, 10, 1265.
2- Materials and Methods
- This section is very well presented and complete. However, quality of figure 1 should be enhanced and references to most of the equations presented should be provided.
Answer to reviewers: Thanks for the comment. The the figure is improved.
3- Results and discussion
1- There are a lot of interesting observations without deep analysis. More physical analysis are to be added to this section;
2- Quality of Figure 1 should be enhanced.
3- Table 1 at the end of page 6 should be Table 2.
- Figure 2 is missing. Please do the necessary updates.
Answer to reviewers: Thanks for the comment. Captions and figures have been revised. We proofread the texts.
4- Conclusions
The main outputs of the manuscript in terms of applications should be highlighted.
Answer to reviewers: Thanks for the comment. The text is improved
5- References
References relevant to Polymers should be added, if possible
Answer to reviewers: Thanks for the comment. New references have been added
Colom, X.; Cañavate, J.; Carrillo-Navarrete, F. Towards Circular Economy by the Valorization of Different Waste Subproducts through Their Incorporation in Composite Materials: Ground Tire Rubber and Chicken Feathers. Polymers 2022, 14, 1090.
Candau, N.; Oguz, O.; León Albiter, N.; Förster, G.; Maspoch, M.L. Poly (Lactic Acid)/Ground Tire Rubber Blends Using Peroxide Vulcanization. Polymers 2021, 13, 1496.
Zedler, Ł.; Klein, M.; Saeb, M.R.; Colom, X.; Cañavate, J.; Formela, K. Synergistic Effects of Bitumen Plasticization and Microwave Treatment on Short-Term Devulcanization of Ground Tire Rubber. Polymers 2018, 10, 1265.
Round 2
Reviewer 3 Report
Thank you for taking into consideration my comments. The manuscript is now ready for publication.